# Safety and Efficacy of Prestige Coils for Embolization of Vascular Abnormalities: The Embo-Prestige Study

**DOI:** 10.3390/jpm13101464

**Published:** 2023-10-05

**Authors:** Julien Frandon, Romaric Loffroy, Clement Marcelin, Hélène Vernhet-Kovacsik, Joel Greffier, Djamel Dabli, Skander Sammoud, Pierre Marek, Olivier Chevallier, Jean-Paul Beregi, Hervé Rousseau

**Affiliations:** 1Department of Medical Imaging, IPI Plateform, Nîmes University Hospital, University of Montpellier, Medical Imaging Group Nîmes, IMAGINE, 30029 Nîmes, France; joel.greffier@chu-nimes.fr (J.G.); djamel.dabli@chu-nimes.fr (D.D.); skander.sammoud@chu-nimes.fr (S.S.); jean.paul.beregi@chu-nimes.fr (J.-P.B.); 2Department of Radiology, CHU Dijon-Bourgogne, 21079 Dijon, France; romaric.loffroy@chu-dijon.fr (R.L.); olivier.chevallier@chu-dijon.fr (O.C.); 3Centre Hospitalier Universitaire de Bordeaux, Service d’Imagerie Diagnostique et Thérapeutique de l’Adulte, Hopital Pellegrin, 33076 Bordeaux, France; clement.marcelin@gmail.com; 4Bordeaux Institute of Oncology, BRIC U1312, INSERM, Université Bordeaux, 33076 Bordeaux, France; 5Department of Radiology, CHU of Montpellier, Arnaud de Villeneuve Hospital, 34090 Montpellier, France; h-vernhet@chu-montpellier.fr; 6Interventional Radiology Department, Rangueil Hospital, University Hospital of Toulouse, 31400 Toulouse, France; marek.p@chu-toulouse.fr (P.M.); rousseau.h@chu-toulouse.fr (H.R.)

**Keywords:** interventional radiology, embolization, peripheral vascular lesions, coils, safety

## Abstract

A wide variety of coils are available for vascular embolization. This study aimed to evaluate the safety and efficacy of a new Prestige coil. We carried out retrospective analysis of a multicenter’s registry data collected between February 2022 and November 2022. The choice of embolization agent used to treat peripheral vascular anomalies was left to the investigator’s discretion. Patients for whom at least one Prestige coil was used were included in Series 1. All other patients were included in Series 2. Efficacy and safety were evaluated. Patients were followed up for one month. In total, 220 patients were included, 110 in each series. Patients included 149 men (67.7%) and 71 women (32.3%), with a median age of 62.5 years (IQR: 35.8–73). Patient ages were similar in the two series. Complete occlusion of the targeted vessel was reported in 96.4% (*n* = 106/110) of patients in Series 1 and in 99.7% (*n* = 109/110) in Series 2. Four patients experienced non-serious adverse events (1.8%, *n* = 4/220): one experienced back pain and one vomiting in Series 1; one patient had off-target embolization and one a puncture site hematoma in Series 2. Sixteen patients (7.2%, *n* = 16/220) were lost to follow up. Improvement in the patient’s general state at one month was reported in 79.0% (*n* = 83/105) of patients in Series 1 and in 74.7% (*n* = 74/99) in Series 2. Ten deaths occurred, five in Series 1 (4.8%, *n* = 5/105) and five in Series 2 (5.1%, *n* = 5/99). These deaths all concerned critically ill patients embolized for emergent arterial bleeding. In conclusion, the 1-month follow-up showed that Prestige coils, alone or in combination, are efficient and safe.

## 1. Introduction

Interventional radiology (IR) has developed rapidly in recent years and, among the IR procedures, embolization is more frequently used and is now part of routine care in various treatments, including treatment of vascular anomalies. These interventions are performed using different embolization agents, either as a single-use intervention or in combination with other agents: coils, resorbable embolization agents (gelatin), liquid agent (cyanoacrylates or ethylene vinyl alcohol copolymers), plugs, or microparticles. Endovascular embolization using metallic coils was first reported in the treatment of vascular malformations such as cranial [1,2] and extra-cranial aneurysms [3]. It is now recognized as an effective, minimally invasive and safe treatment.

A wide variety of coils are available for embolization of vascular lesions. These coils are of different lengths, diameters, shapes, stiffnesses and types, including fibrous or hydrogel-coated coils, with different types of detachment: controlled mechanical detachment coils or pushable coils. The coils each have advantages and disadvantages, but have been shown to be equally effective in terms of vascular occlusion [4,5,6]. Coils can be thick and highly occlusive but rigid, requiring a large-diameter microcatheter for placement, or thin and flexible, sometimes requiring multiple coils to achieve a dense packing, with greater risk of compaction and permeabilization of the aneurysm [7,8]. Hydrogel-coated coils have been developed to be more occlusive. Studies have shown these to be safe to use [9] and more effective in occluding than conventional/naked coils [10,11]. Some coils can be released very simply by pushing them out of the catheter, but to ensure greater precision in the procedure, controlled-release systems have been developed [12]. Among these, mechanical release coils are hooked to the tip of the pusher, requiring a microcatheter large enough to contain the delivery system. Other controlled-release systems based on an electrical process allow the use of smaller microcatheters [1]. Even with the same structural composition, each type of coil is unique in terms of mechanical properties, and they show different degrees of flexibility/stiffness, specific deployment configurations and variations in packing density. It is therefore important to evaluate new coils in terms of safety and efficacy in the embolization treatment of vascular anomalies.

Prestige coils (Balt, Montmorency, France) have recently been developed as platinum-based radiopaque coils, with an electrical detachment system that allows the utilization of large-volume coils through a very small microcatheter. Both complex and helical coils are offered and these range in size from 2 mm to 24 mm and in length from 2 cm to 65 cm, all compatible with 0.0163″–0.022″ microcatheters. This type of platinum-based coil was initially used for the treatment of intracranial aneurysms [13] and was later employed for the embolization of various peripheral vascular lesions [10]. The main objectives of this multicenter retrospective study were to evaluate the efficacy and safety of Prestige coils and other materials used in peripheral-vessel embolization, and to describe the indications and characteristics of the use of these Prestige coils, as well as to record the data related to the use of these coils.

## 2. Patients and Methods

### 2.1. Study Design and Patients

The study retrospectively analyzed data of 220 patients prospectively collected during standard care. The study was conducted in accordance with the Declaration of Helsinki. It was performed according to the MR004 methodology, according to 2016–41 law dated 26 January 2016 on the modernization of the French health system, and approved by the Institutional Review Board of the Nîmes University Hospital (IRB number 22-11-01). Informed consent was obtained from all subjects involved in the study.

Patients included in the study were those 18 years or older presenting vascular lesions outside neuroradiology, with indication for vessel occlusion using arterial or venous coils, scheduled or emergency. Patients were excluded if they had had previous implantation of a Prestige coil in the same area, or if they had participated in another study or were currently in the exclusion period of a previous study.

Patients were divided into two series: the Prestige series (Series 1), with patients who underwent embolization with the use of Prestige coils, alone or in combination with other embolization materials (other coils or other materials) and the control series (Series 2), which included all other patients who underwent embolization without the use of a Prestige coil, whatever other embolization materials used. All data are available upon reasonable request.

### 2.2. Embolization Agents and Procedure

Prestige coils are platinum–tungsten coils with a steel hypotube and a controlled-detachment system, and are indicated for peripheral vascular embolization. Prestige coils can be used alone or in combination with other embolization agents. Other embolization agents include other coils, liquid agent, spongel, microparticles, or plugs.

The embolization procedure was performed by an interventional radiologist. The choice of device (Prestige coil, other coil or other embolization material) was left to the discretion of the operator, according to their usual practice. The approach was femoral, humeral or radial. Selective catheterization with the carrier catheter was performed, followed by microcatheterization of the target vessel. The correct positioning of the microcatheter was checked by injection of contrast medium before the embolization material was placed. A final control of the good occlusion of the vascular anomaly was systematically performed.

### 2.3. Follow-Up

Follow-up were performed at 1 month, with a follow-up visit or by telephone according to the center’s standard practice. Clinical success, the patient’s general condition and possible adverse events were collected during each follow-up.

### 2.4. Study Endpoints

The primary efficacy endpoint was successful embolization, defined as complete occlusion of the target vessel(s) with no residual filling of the embolized lesion(s), measured by angiography immediately after the procedure. The clinical success, evaluated at 1 month, was defined as the absence of (1) clinical worsening, (2) reintervention (surgery for a second embolization) for the embolized target lesion or (3) death due to hemorrhagic recurrence within the first month after the procedure. The patient’s general condition (improvement, return to baseline state, stationary state, worsening) was assessed during the patient follow-up. The technical success was defined as the correct coil or plug deployment in cases where coils were used (Prestige or other coils and plugs), or as good ballistics of the embolization material (absence of uncontrolled reflux) in cases where a liquid agent, spongel or microparticles were used. The technical risk of embolization was assessed before the start of the procedure by the radiologist, using a 4-point Likert scale “I consider the risk of the embolization procedure to be: 1, low; 2, medium; 3; high or 4, very high”. The technical characteristics of Prestige coils—flexibility, pushability and packing density—were evaluated using 4-point Likert scales. The radiologist’s confidence in the procedure was assessed immediately after the procedure, using a 4-point Likert scale, from 1: very low confidence to 4: very confident.

### 2.5. Statistical Analysis

Given the exploratory nature of this study, a formal calculation of the required sample size was not performed. However, to ensure an adequate sample size for studying the primary objective, it was determined that the inclusion of 100 patients per series would be appropriate. To account for potential loss to follow-up at the 1-month mark, an additional 10% of patients were included in each series. Patients were prospectively and consecutively enrolled until the predetermined target number of inclusions was reached in each series.

Statistical analyses were descriptive, performed using biostattgv.com. Quantitative variables are presented with medians and 1st and 3rd quartiles (IQR), and categorical variables with numbers of patients and percentages.

## 3. Results

### 3.1. Patients

A total of 220 patients were included in the study in five participating centers, 110 in each series (Figure 1 flow chart). Of these, 204 patients completed their one-month follow-up.

Patients included 149 men (67.7%) and 71 women (32.3%), with a median age of 62.5 years (IQR: 35.8–73). Patient ages were similar in the two series (Table 1). Patients were mostly of performance status 0 (*n* = 115, 52.8%) or 1 (*n* = 62, 28.4%). Patients in the Prestige series reported more comorbidities (diabetes, arterial hypertension or cirrhosis, *p* = 0.011) and previous anti-coagulant and anti-platelet treatments (*p* = 0.006) than patients in the control series.

### 3.2. Embolization Data

Before the embolization procedure, the interventional radiologist was asked to rate the estimated risk of the procedure as poor, medium, high or very high. The procedure was rated as low risk in 35.5% of cases, and as high risk in 23.6% (Table 2).

Most embolizations (165, 75%) were performed on arterial anomalies, with 52 (23.6%) being performed on venous anomalies. The median size of the anomaly was 6 mm (IQR: 4–12), and the median size of the occluded vessel was 5 mm (IQR: 3–6), higher in the control than in the Prestige series (Table 2).

Prestige coils were used in 110 patients (Series 1, 50%). They were used alone in 68/110 patients (61.8%) and in combination with other embolization agents in 42 (38.2%) of cases. Other embolization agents used in Series 1 were coils other than Prestige coils in 16 patients (14.5%), a liquid agent in 24 patients (21.8%), spongel in 6 patients (5.5%), microparticles in 3 patients (2.7%) and plugs in 3 patients (2.7%) (Table 3).

In Series 2, 49 patients (44.5%) were embolized with coils, 50 patients (45.5) with a liquid agent, 16 (14.5%) with spongel, 7 (6.4%) with microparticles and 16 (14.5%) with plugs.

Among patients treated with coils, whatever the series, with Prestige or other coils, fewer than 10 coils were pushed in 134/159 patients (84.8%); the first coil pushed was of diameter < 6 mm for 91/159 patients (58.0%), and of length ≤ 15 cm for 89/159 patients (56.3%). A 2.7fr microcatheter was used in 123/210 patients (58.6%). Large microcatheters (2.7fr) were less frequently used in the Prestige series (*n* = 43/107, 40.2%) as compared to the control series (*n* = 80/203, 77.7%, *p* < 0.001). The scopy time was 25 min (IQR: 16–39) in the Prestige series, significantly longer than in the control series (17.5 min (IQR: 10–30), *p* = 0.015).

### 3.3. Prestige Coils Characteristics

Data on the use of Prestige coils are presented in Table 4. The majority of Prestige coils were considered flexible or very flexible in all procedures and very easy or easy to push in 98.2% of procedures, and the packing density obtained was very dense or quite dense in more than 80% of procedures.

### 3.4. Immediate Efficacy

Immediate clinical efficacy was evaluated at the end of the procedure. The complete occlusion of the targeted vessel, the primary endpoint of the study, was reported in 96.4% (*n* = 106/110) of patients in the Prestige series and in 99.7% (*n* = 109/110) in Series 2, so it was not significantly different in the two series (Table 5). In the Prestige group, the coils were correctly positioned in all 110 patients.

### 3.5. One-Month Efficacy

Short-term follow-up was collected for patients from the two series (Table 6). Most patients had a better general state than before the intervention, 79% in the Prestige group and 74.7% in the control group. Among all patients, 10 patient (5.4%) died before the 1-month follow-up.

The median delay for the one-month follow-up (data available for 162/220 patients) was 34 days (IQR: 30–45). Of these, 48/162 follow-ups were performed within the first month (<30 days), and 111/162 within 40 days.

### 3.6. Safety

Four (4) non-serious adverse events without clinical consequences were declared in four patients (1.8%, *n* = 4/220), including: one vomiting and one back pain not related to the device in the Prestige group (Series 1); and one migration of material during embolization of complex vascular malformations which was related to the device and one hematoma at the punction site which were not related to the device in Series 2 [14].

Three (3) device deficiencies/technical complications without clinical consequences occurred in three patients (1.4%, *n* = 3/220); all occurred in Series 1 (2.7%, *n* = 3/110). Two of these complications corresponded to coils that were difficult to push, which could be explained by a handling error resulting in a failure to detachment (one patient) or an early detachment of the coil inside the microcatheter (one patient).

Ten (10) deaths were reported at the 1-month follow up, five occurred in the Prestige group (5.7%, *n* = 6/105) and five in the other group (5.1%, *n* = 5/99), *p* = 0.92. They all concerned patients embolised for arterial bleeding with a good outcome of the embolization at the end of the procedure.

## 4. Discussion

This study was the first multicentric study conducted with more than 200 patients to assess the safety and efficacy of Prestige coils compared to other embolization materials used in clinical practice. The study gives an overview of embolization of vascular anomalies in clinical practice in expert centres in France. The study also provides safety and immediate efficacy data of real-life clinical practice of embolization with and without Prestige coils in five French centers and the 1-month follow-up after embolization.

Our results showed that in 220 patients Prestige coils used in the context of embolization of peripheral vascular anomalies were safe and efficient. Efficacy of the embolization procedure, i.e., complete occlusion of the target vessel, was similar in the two series (96.4% in the Prestige series and 99.1% in the control series), and the Prestige coils were reported as correctly positioned (technical success) in all cases (110/110 patients). Our results show that Prestige coils, alone or in combination with other embolization agents (other coils, liquid agents, plugs…), are efficient when used for embolization of peripheral vascular anomalies, in various indications and contexts (scheduled or emergency), in various patients in everyday practice.

Very few studies have yet been published on the use of Prestige coils. Our results however are concordant with one previously published study [15]. The coils were shown to be efficient in hemorrhoid embolization requiring soft microcoils and small catheters to obtain occlusion of small arteries [15]. In this study, technical success was obtained in 100% (21/21) of the cases. Three (14.2%) patients underwent a second embolization due to rebleeding. One patient (4.7%) underwent surgery. No major complications were observed. Three patients had minor complications (one case of radial hematoma and two cases of minor tenesmus). We also report very good safety. Among the 110 patients in the Prestige series, 5 deaths and 2 non-serious AE were reported, not related to the device used. The deaths concerned severe patients with either significant comorbidities or in haemorrhagic shock, treated for arterial anomalies. These deaths were not related to the device or the embolization procedure. It is difficult to compare with the literature because this was a real-life study, without selection of patients, and included patients taken in emergency for multivisceral distress. There were two device deficiencies (detachment problems) and one technical complication (movement of the coil) which had no clinical consequence. This may have been due to the fact that these coils were new to the interventional radiologists participating in the study. It is likely that a learning curve is required to get used to the detachment of these new coils. However, operators were more often very confident with Prestige coils than with other embolic materials (50.0% vs. 35.5%, *p* = 0.04) which shows that once they were used to this new material they were comfortable with it.

Prestige coils seem more adequate for small vessels and allow the use of small micro-catheters, potentially due to the miniaturized electric detachment system. This may be of use with the generalization of the embolization technique in other and newly-spreading indications such as prostatic artery [16] or shoulder embolization [17] in which use of a microcatheter is essential in the small arteries implicated in these pathologies. Prestige coils were found to be flexible, but packing was not very tight in our study, as compared to a previous study by Hongo et al. which highlighted the packing density provided by hydrocoils [10]. However, the efficacy was similar to that of the control series, despite the fact that patients in the Prestige series had more frequently had anticoagulant treatments.

Hydrocoils represent a newer generation of coils coated with bioactive polymers, showcasing enhanced occlusive properties but without a proven definitive clinical advantage [18]. These coils were developed to address the occlusive limitations observed in older generation of bare platinum coils [10]). However, hydrocoils tend to be larger and less flexible (requiring at least a 0.021” microcatheter according to Hongo et al.). This may limit their utility in complex cases with unstable catheterization. Consequently, platinum coils must adapt to confront these emerging challenges associated with flexibility and packing density. Recent studies have showed the flexibility of new generation of bare platinum coils, striking a balance between flexibility due to their uncoated design and robust packing capability [19,20]. Indeed, coil flexibility enables better coiling and packing of vascular anomalies, even in complex access scenarios. Prestige coils align with this trajectory, contributing to the ongoing evolution of coil technology to address these demands.

Due to the absence of randomization, it is important to note that patient populations treated with Prestige coils and other types of coils exhibited different characteristics. Prestige coils were more frequently used in patients with comorbidities and those receiving antithrombotic treatment prior to the procedure. Additionally, there was a tendency to use Prestige coils more frequently in the treatment of arterial lesions. Prestige coils have proven to be highly flexible and suitable for complex vascular anatomies as well as small vessels. As a result, operators may have chosen them as a priority for more complex or higher-risk procedures. Despite these initial differences, follow-up results demonstrated similar efficacy between the two coil groups, suggesting that Prestige coils were effective even in more complex scenarios.

Hongo et al. showed a lower number of coils and a shorter length with hydrogen-coated coils than with the non-hydrogen-coated coils [10]. In our study, there was no difference in the number of coils used in the two groups. The total length was not measured.

The fluoroscopy time was longer in the Prestige series than in the control series. This may be explained by the presence of smaller arteries, or more complex procedures requiring smaller microcatheter (*p* < 0.001). However, there was no impact on the patients’ radiation dose thanks to the dose optimization work performed in interventional radiology suites, as shown in a previous published study on interventional radiology practices [21].

This study had some limitations, including the study design without randomization and the lack of an independent core lab review. Its retrospective nature could have influenced the strength of the evidence. Also, packing density was not precisely calculated. The populations were heterogeneous but this study reflects real-life clinical practices of interventional radiology departments in five French centers.

## 5. Conclusions

This multicenter study, involving 220 patients, demonstrated the safety and effectiveness of Prestige coils when compared to other embolization materials commonly used in clinical practice. The results were positive both immediately after the procedure and at the one-month follow-up. This study gives an overview of embolization of vascular anomalies in clinical routine in five expert centres in France. Prestige coils exhibited efficacy when used in conjunction with other materials as well as when employed independently. They were found to be suitable for various indications, artery types and sizes. This diversity enhances the options available for personalized medicine, allowing for the selection of one or more embolization agents based on the specific indication and pathology.

## Figures and Tables

**Figure 1 jpm-13-01464-f001:**
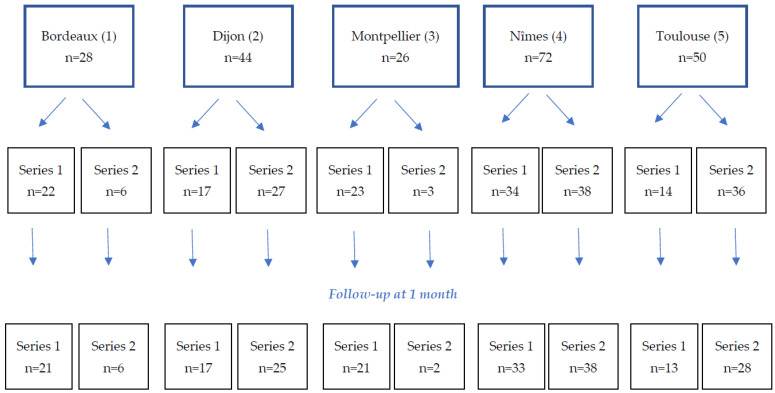
Flow chart detailing inclusions in the five centers and the one-month follow-up.

**Table 1 jpm-13-01464-t001:** Patients’ characteristics at baseline.

*n* (%)	Series 1 (*n* = 110)	Series 2 (*n* = 110)	Total (*n* = 220)	*p*-Values
Age *	64.5 (38.3–73.0)	59.0 (34.0–71.8)	62.5 (35.8–73)	0.13
Gender				
Male	71 (64.5)	78 (70.9)	149 (67.7)	0.31
Female	39 (35.5)	32 (29.1)	71 (32.3)	
Comorbidities				** *0.011* **
Diabetes	13 (11.7)	7 (6.4)	20 (9.1)	
Arterial hypertension	47 (42.3)	31 (28.2)	78 (35.5)	
Cirrhosis	4 (3.6)	2 (1.8)	6 (2.7)	
Performance status				0.14
0	50 (45.9)	65 (59.6)	115 (52.8)	
1	37 (33.9)	25 (22.9)	62 (28.4)	
2	14 (12.8)	15 (13.8)	29 (13.3)	
3	3 (2.8)	3 (2.8)	6 (2.8)	
4	5 (4.6)	1 (0.9)	6 (2.8)	
Missing	*1*	*1*	*2*	
Previous treatments				** *0.006* **
Anti-coagulant	24 (21.6)	11 (10.0)	35 (15.9)	
Anti-platelet	25 (22.5)	16 (14.5)	41 (18.6)	

* This variable is expressed with its median and interquartile range (IQR). Significant *p*-values (*p* < 0.005) are presented in italics and bold.

**Table 2 jpm-13-01464-t002:** Vascular anomaly data.

*n* (%)	Series 1 (*n* = 110)	Series 2 (*n* = 110)	Total (*n* = 220)	*p*-Values
Risk of the procedure				0.16
Low	42 (38.2)	36 (32.7)	78 (35.5)	
Medium	48 (43.6)	42 (38.2)	90 (40.9)	
High	20 (18.2)	32 (29.1)	52 (23.6)	
Vascular anomaly				0.08
Arterial	89 (80.9)	78 (70.0)	167 (75.9)	
Venous (variquous)	20 (18.2)	32 (29.1)	52 (23.6)	
Other	1 (0.9)	0	1 (0.05)	
Arterial anomalies				0.21
Aneurism	21 (23.6)	10 (14.3)	31 (21.1)	
Faux aneurism	18 (20.2)	19 (27.1)	37 (25.2)	
Bleeding	29 (32.6)	36 (51.4)	65 (44.2)	
Arteriovenous fistula	4 (4.5)	3 (4.3)	7 (4.8)	
Malformation	5 (5.6)	2 (2.9)	7 (4.8)	
Missing	*12*	*8*	*19*	
Embolized artery				
Hepatic artery	3 (3.5)			
Splenic artery	13 (15.1)			
Superior mesenteric artery	4 (4.7)			
Inferior mesenteric artery	4 (4.7)	NA	NA	
Left gastric artery	1 (1.2)			
Gastroduodenal artery	5 (5.8)			
Hypogastric artery	13 (15.1)			
Renal artery	13 (15.1)			
Other	30 (34.9)			
Missing	*3*			
Embolized vein				
Spermatic vein	16 (80.0)			
Ovarian vein	1 (5.0)	NA	NA	
Other	3 (15.0)			
Size of the vascular anomaly *	6 (4–14)	6 (5–10)	6 (4–12)	0.056
Missing	*0*	*1*	*1*	
Size of the occluded vessel *	4 (3–6)	5 (3–6)	5 (3–6)	** *0.038* **

* This variable is expressed with its median and interquartile range (IQR). Significant *p*-values (*p* < 0.005) are presented in italics and bold. NA, not applicable; data not collected for Series 2.

**Table 3 jpm-13-01464-t003:** Embolization technical data.

*n* (%)	Series 1 (*n* = 110)	Series 2 (*n* = 110)	Total (*n* = 220)	*p*-Values
Embolization agent used				
Prestige coils	110 (100)	0	110 (50)	
Other coils	16 (14.5)	49 (44.5)	65 (29.5)	
Liquid agent	24 (21.8)	50 (45.5)	74 (33.6)	
Spongel	6 (5.5)	16 (14.5)	22 (10.0)	
Microparticles	3 (2.7)	7 (6.4)	10 (4.5)	
Plug	3 (2.7)	16 (14.5)	19 (8.6)	
Number of coils used **				0.11
0–9	89 (80.9)	45 (93.8)	134 (84.8)	
10–29	17 (15.5)	3 (6.3)	20 (12.7)	
30–50	4 (3.6)	0	4 (2.5)	
Missing	*0*	*1*	*1*	
Length of the first coil pushed **				0.4
≤15 cm	63 (57.3)	26 (54.2)	89 (56.3)	
15–30 cm	27 (24.5)	9 (18.8)	36 (22.8)	
>30 cm	20 (18.2)	13 (27.1)	33 (20.9)	
Missing	*0*	*1*	*1*	
Diameter of the first coil pushed **				0.31
<6 mm	65 (59.1)	26 (55.3)	91 (58.0)	
6–10 mm	27 (24.5)	17 (36.2)	44 (28.0)	
11–20 mm	13 (11.8)	2 (4.3)	15 (9.6)	
≥20 mm	5 (4.5)	2 (4.3)	7 (4.5)	
Missing	*0*	*2*	*2*	
Microcatheter size				** *<0.001* **
≤2.0 fr	12 (11.2)	5 (4.9)	17 (8.1)	
2.4 fr	52 (48.6)	18 (17.5)	70 (33.3)	
≥2.7 fr	43 (40.2)	80 (77.7)	123 (58.6)	
Missing	*3*	*7*	*10*	
Scopy time *	25 (16–39)	17.5 (10–30)	22 (13–37)	** *0.015* **
Missing	2	10	12	
Radiation dose *	47,060 (23,637–115,621)	34,385 (18,077–98,077)	39,470 (19,550–107,878)	0.26
Missing	*3*	*10*	*10*	

* This variable is expressed with its median and interquartile range (IQR). ** Percentages are calculated on the number of patients treated with a coil. Significant *p*-values (*p* < 0.005) are presented in italics and bold.

**Table 4 jpm-13-01464-t004:** Characteristics of Prestige coils.

*n* (%)	Series 1 (*n* = 110)
Flexibility of the coil	
Very flexible	61 (55.5)
Flexible	49 (44.5)
Rigid	0
Very rigid	0
Pushability	
Very easy to push	61 (55.5)
Easy to push	47 (42.7)
Difficult to push	2 (1.8)
Very difficult to push	0
Packing density	
Not dense at all	2 (1.8)
A little dense	17 (15.6)
Quite dense	56 (51.4)
Very dense	34 (31.2)
Missing	*1*

**Table 5 jpm-13-01464-t005:** Efficacy of the embolization procedure according to the embolization agent used, in the two series.

*n* (%)	Series 1 (*n* = 110)	Series 2 (*n* = 110)	Total (*n* = 220)	*p*-Values
Correct positioning of the coil/embolization agent				
Prestige coil	110 (100)	NA		
Other coils *	15 (13.6)	46 (41.8)		
Liquid agent *	24 (21.8)	49 (44.5)		
Spongel *	6 (5.5)	15 (13.6)	NA	
Microparticles *	2 (1.8)	7 (6.4)		
Plug *	3 (2.7)	15 (13.6)		
Complete occlusion of the target vessel	106 (96.4)	109 (99.1)	215 (97.7)	0.17
Confidence of the radiologist in his/her procedure				** *0.04* **
Not very confident	1 (0.9)	0	1 (0.5)	
A bit confident	1 (0.9)	1 (0.9)	2 (0.9)	
Confident	53 (48.2)	70 (63.6)	123 (55.9)	
Very confident	55 (50.0)	39 (35.5)	94 (42.7)	

* Percentages were calculated on the number of patients treated with the given treatment. Significant *p*-values (*p* < 0.005) are presented in italics and bold. NA, not applicable.

**Table 6 jpm-13-01464-t006:** One-month efficacy follow-up.

*n* (%)	Series 1 (*n* = 105)	Series 2 (*n* = 99)	Total (*n* = 204)	*p*-Values
Lost to follow-up/missing	5	11	16	0.12
Improvement	83 (79.0)	74 (74.7)	157 (77.0)	0.99
Steady state	1 (1.0)	1 (1.0)	2 (1.0)	0.97
Back to baseline state	16 (15.2)	18 (18.2)	34 (16.7)	0.57
Aggravation Incl death	6 (5.7) 5 (4.8)	6 (6.1) 5 (5.1)	12 (5.9) 10 (5.4)	0.92

## Data Availability

All data are available upon reasonable request.

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
