# Peer review of "Safety and Efficacy of Prestige Coils for Embolization of Vascular Abnormalities: The Embo-Prestige Study"

_jpm, 2023, doi:10.3390/jpm13101464_

Round 1
Reviewer 1 Report
Dear Authors!
Thank you for the interesting data presented.
I have some minor remarks.
Title. I believe prestige should have capital P.
Line 44. EVOH has to be written fully before abbreviation is used.
Lines 69-71. This statement needs references.
Line 79. Please add a reference about MR004.
Paragraph with lines 92-98 – end-points have to be described first, before sample size is reported.
Sub-section 2.2 is unnecessary and has to be removed as study end-points are reported below.
Please, provide an explanation of the possible bias related to differences in patients’ conditions. What is your opinion why patients in Prestige coils group had more comorbidities, were more frequently prescribed with antithrombotic treatment before? Why there was a tendency to use Prestige coils in arterial lesions more frequently? All this has to be discussed.
On my opinion the main limitation is retrospective design. This has to be stated clearly.
Conclusions have to be rewritten. Two first statements are not study conclusions at all. Please, be more precise and report only what related to what have been found, i.e. Prestige coils are safe and efficient, etc.
None
Author Response
Reviewer 1:
Thank you for your thoughtful and constructive feedback on our manuscript. In this response, we will address each of your comments and suggestions point by point to enhance the quality and clarity of our work.
I have some minor remarks.
R1C1: Title. I believe prestige should have capital P.
We corrected this tippo.
R1C2: Line 44. EVOH has to be written fully before abbreviation is used.
EVOH was written fully.
R1C3: Lines 69-71. This statement needs references.
This sentence has been changed because it was confusing. In fact, this is the first study on prestige coils. We have therefore modified and added references as requested by the reviewer: “This type of platinum-based coil was initially used for the treatment of intracranial aneurysms [add: PMID: 23074479] and was later employed for the embolization of various peripheral vascular lesions [add ref 10: hongo et al]
R1C4: Line 79. Please add a reference about MR004.
There isn't a specific reference; it's a legal requirement. We have clarified as follows: “It was performed according to the MR004 methodology, according to 2016–41 law dated 26 January 2016 on the modernisation of the French health system, and approved by the Institutional Review Board of the Nîmes University Hospital (IRB number 22-11-01)”. Is that acceptable to you?
R1C5: Paragraph with lines 92-98 – end-points have to be described first, before sample size is reported.
The paragraph has been moved to the end of the method section, in the statistics section.
R1C6: Sub-section 2.2 is unnecessary and has to be removed as study end-points are reported below.
We removed this part.
R1C7: Please, provide an explanation of the possible bias related to differences in patients’ conditions. What is your opinion why patients in Prestige coils group had more comorbidities, were more frequently prescribed with antithrombotic treatment before? Why there was a tendency to use Prestige coils in arterial lesions more frequently? All this has to be discussed.
Thank you for your comment. We have added a paragraph to the discussion: " Due to the absence of randomization, it was important to note that patient populations treated with Prestige coils and other types of coils exhibited different characteristics. Prestige coils were more frequently used in patients with comorbidities and those receiving antithrombotic treatment prior to the procedure. Additionally, there was a tendency to use Prestige coils more frequently in the treatment of arterial lesions. Prestige coils have proven to be highly flexible and suitable for complex vascular anatomies as well as small vessels. As a result, operators may have chosen them as a priority for more complex or higher-risk procedures. Despite these initial differences, follow-up results demonstrated similar efficacy between the two coil groups, suggesting that Prestige coils were effective even in more complex scenarios.”
R1C8: On my opinion the main limitation is retrospective design. This has to be stated clearly.
We emphasized this limitation as suggested: “Its retrospective nature could have influenced the strength of the evidence”
R1C9: Conclusions have to be rewritten. Two first statements are not study conclusions at all. Please, be more precise and report only what related to what have been found, i.e. Prestige coils are safe and efficient, etc.
Conclusions was rewritten: “This multicenter study, involving 220 patients, demonstrated the safety and effectiveness of Prestige coils when compared to other embolization materials commonly used in clinical practice. The results were positive both immediately after the procedure and at the one-month follow-up . The complete occlusion of the targeted vessel was reported in 96.4% and improvement of the patient’s general state at one month was reported in 79.0%. This study gives an overview of embolization of vascular anomalies in clinical routine in 5 expert centres in France. Prestige coils exhibited efficacy when used in conjunction with other materials as well as when employed independently. They were found to be suitable for various indications, artery types, and sizes. This diversity enhances the options available for personalized medicine, allowing for the selection of one or more embolization agents based on the specific indication and pathology.”
Reviewer 2 Report
The paper constitutes an interesting and well-prepared work. The research subject is definitely worth investigating. The only minor revisions suggested to Authors are as follows:
1) Words such as Background, Method, Results and Conclusion should be removed from abstract of the paper.
2) Section 2.1.: information concerning the MR004 methodology should be supported by adequate literature reference.
3) Section 3.1.: There is no adequate Figure caption below the graph showing the results of the research.
4) Discussion over the results should be supported to a more extent by references to other works.
5) Final conclusions should be supplemented with some quantified data.
6) Section References should be significantly extended including more up-to-date references and prepared in line with the requirements of the Journal.
The paper contains some grammar or linguistic mistakes hence it should be re-checked.
Author Response
Reviewer 2
We appreciate your feedback regarding our manuscript. In this response, we will carefully address each of your comments and recommendations individually, with the aim of improving the overall quality and clarity of our work
The paper constitutes an interesting and well-prepared work. The research subject is definitely worth investigating. The only minor revisions suggested to Authors are as follows:
R2C1 Words such as Background, Method, Results and Conclusion should be removed from abstract of the paper.
Done.
R2C2 Section 2.1.: information concerning the MR004 methodology should be supported by adequate literature reference.
We have clarified as follows: “It was performed according to the MR004 methodology, according to 2016–41 law dated 26 January 2016 on the modernisation of the French health system, and approved by the Institutional Review Board of the Nîmes University Hospital (IRB number 22-11-01)”. Is that acceptable to you?
R2C3 Section 3.1.: There is no adequate Figure caption below the graph showing the results of the research.
We modified as: “detailing inclusions in the 5 centres and the one-month follow-up”
R2C4 Discussion over the results should be supported to a more extent by references to other works.
In accordance with your comment, we have added a paragraph to the discussion section as follows: 'Hydrocoils represent a newer generation of coils coated with bioactive polymers, showcasing enhanced occlusive properties but without a proven definitive clinical advantage [doi:10.3390/brainsci12081062]. These coils were developed to address the occlusive limitations observed in older generation of bare platinum coils [Hongo]. However, hydrocoils tend to be larger and less flexible (requiring at least 0.021" microcatheter in Hongo et al). This may limit their utility in complex cases with unstable catheterization. Consequently, platinum coils must adapt to confront these emerging challenges associated with flexibility and packing density. Recent studies have showed the flexibility of new generation of bare platinum coils, striking a balance between flexibility due to their uncoated design and robust packing capability [doi:10.1186/s12883-020-1623-9, doi:10.1016/j.jvscit.2021.01.005]. Indeed, coil flexibility enables better coiling and packing of vascular anomalies, even in complex access scenarios. Prestige coils align with this trajectory, contributing to the ongoing evolution of coil technology to address these demands”.
R2C5 Final conclusions should be supplemented with some quantified data.
We modified the conclusion and added quantified data as follows: “This multicenter study, involving 220 patients, demonstrated the safety and effectiveness of Prestige coils when compared to other embolization materials commonly used in clinical practice. The results were positive both immediately after the procedure and at the one-month follow-up . The complete occlusion of the targeted vessel was reported in 96.4% and improvement of the patient’s general state at one month was reported in 79.0%. This study gives an overview of embolization of vascular anomalies in clinical routine in 5 expert centres in France. Prestige coils exhibited efficacy when used in conjunction with other materials as well as when employed independently. They were found to be suitable for various indications, artery types, and sizes. This diversity enhances the options available for personalized medicine, allowing for the selection of one or more embolization agents based on the specific indication and pathology.”
R2C6 Section References should be significantly extended including more up-to-date references and prepared in line with the requirements of the Journal.
We have taken your advice and expanded our discussion with more references. On the editable version, sent by the journal, we don't have access to the bibliography, so I've added the DOIs of the articles to be added. Similarly, the bibliography has been updated directly by the journal.